# Taxonomy Expansion for Named Entity Recognition

**Karthikeyan K**[1*]**, Yogarshi Vyas**[2†]**, Jie Ma**[2]**, Giovanni Paolini**[2]**, Neha Anna John**[2]**,**
**Shuai Wang**[2]**, Yassine Benajiba**[2]**, Vittorio Castelli**[2]**, Dan Roth**[2]**, Miguel Ballesteros**[2]

[1]Department of Computer Science, Duke University
[2]AWS AI Labs
karthikeyan.k@duke.edu,
{yogarshi,jieman,paoling,nehajohn,wshui,
benajiy,vittorca,drot,ballemig}@amazon.com

## Abstract

Training a Named Entity Recognition (NER) model often involves fixing a taxonomy of entity types. However, requirements evolve and we might need the NER model to recognize additional entity types. A simple approach is to re-annotate entire dataset with both existing and additional entity types and then train the model on the re-annotated dataset. However, this is an extremely laborious task. To remedy this, we propose a novel approach called **P**artial **L**abel **M**odel (PLM) that uses only partially annotated datasets. We experiment with 6 diverse datasets and show that PLMconsistently performs better than most other approaches (0.5–2.5 F1), including in novel settings for taxonomy expansion. The gap between PLM and other approaches is especially large in settings where there is limited data available for the additional entity types (as much as 11 F1), thus suggesting a more cost effective approach to taxonomy expansion.

## 1 Introduction

Training a Named Entity Recognition (NER) model typically involves presupposing a fixed taxonomy, that is, the entity types that are being recognized are known and fixed. However, NER models often need to extract new entity types of interest appearing in new text, thus requiring an expansion of the taxonomy under consideration. Consider an example (Figure 1), where an NER model was initially trained to recognize PERSON and LOCATION types using data annotated only for these two types. However, it later needs to identify ORGANIZATION entities in addition to the two initial entity types.

There are several simple ways to solve this problem, yet they are cumbersome in different ways. One possible solution is to re-annotate all data that was only annotated for the initial entity types (PERSON, LOCATION) with the new entity types

---

*Work done during an internship at AWS AI Labs
†Corresponding author

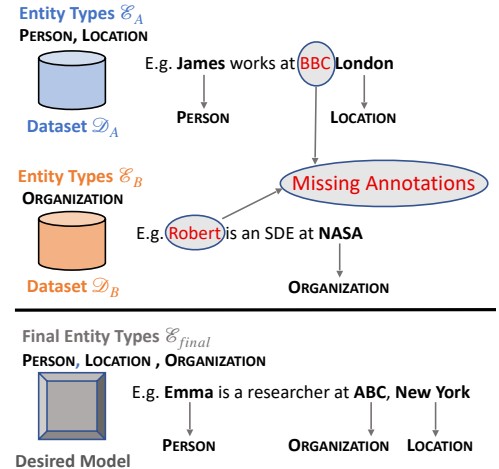

Figure 1: **Taxonomy Expansion for NER:** Given Dataset $\mathcal{D}_A$ annotated only with entity types $\mathcal{E}_A$ and $\mathcal{D}_B$ annotated only with $\mathcal{E}_B$, the task is to train a model that recognize entities from both $\mathcal{E}_A$ and $\mathcal{E}_B$.

(ORGANIZATION), and then train a single model to recognize all the entity types on this data. If such re-annotation and re-training needs to happen iteratively every time new entities are introduced, then this is an expensive process, and re-annotation requires knowledge of all entity types both old and new. A secondary issue with this approach is that the original training data was chosen only to contain examples of the initial entity types. Thus, it may not even contain enough examples of the new entity types and it may be simpler to obtain data that is rich in the new entities types for annotation (ORGANIZATION). Given these two disjoint datasets with two separate sets of annotations, we can train two different models to recognize the two sets of entity types — yet this requires heuristics to combine the predictions of the two models, especially if the old and new entity types have some shared semantics (e.g. some of the new entity types are subtypes of the existing entity types).

This problem of Taxonomy Expansion for NER

(henceforth, TE-NER) has been studied in several different settings in prior work. However, most prior work (§2) assumes that the old and the new entity types are completely disjoint and mutually exclusive. In contrast, we define a general version of the TE-NER problem (§3) where the original training data can be used, and old and new entity types can be related in different ways. Our general setup allows for many practical scenarios encountered during taxonomy expansion such as new entity types being subtypes of existing entity types, or partially overlapping in definition.

We then propose a novel solution to TE-NER motivated by the fact that the available training data are only partially annotated (§4.2). As shown in Figure 1, **BBC** has not been annotated as ORGANIZATION in the initial dataset $\mathcal{D}_A$, since ORGANIZATION was not in the label taxonomy of $A$. Similarly, **Robert** is not annotated as PERSON in the new dataset $\mathcal{D}_B$. Given such partial annotations, we treat the true, unobserved label as a latent variable and derive our approach called **P**artial **L**abel **M**odel (PLM). Intuitively, PLM uses a model trained on $\mathcal{D}_B$ to annotate **BBC** with a probability distribution over entity types $B$. Similarly, it uses a model trained on $\mathcal{D}_A$ to obtain predictions for **Robert**. Finally, the desired model over all entity types is trained using both the observed annotation and the soft labels obtained via the predictions of the intermediate models. Minimizing the KL divergence loss (as used by Monaikul et al. (2021) and Xia et al. (2022)) corresponds to maximizing a lower bound on the likelihood of our formulation, while the proposed PLM loss corresponds to directly maximizing the likelihood.

Experiments with six English datasets show that PLM outperforms all baselines except one (cross annotation) by 0.5–2.5 F1 (§6). However, this competitive baseline, that involves simply *cross-annotating* the datasets using weak labels from a model trained on the other dataset, is less accurate in few-shot settings compared to PLM. When only 200 examples are provided for each new entity type, PLM is 11 F1 better than cross-annotation. We further test the robustness of these approaches across several data distributions and find that they perform robustly across the board (§7).

## 2 Related Work

While this work is closely related to knowledge distillation (Hinton et al., 2015; Shmelkov et al.,

2017) and continual learning (CL) based methods for NER (Xia et al., 2022; Monaikul et al., 2021; Chen and Moschitti, 2019; De Lange et al., 2019), our setup differs in two aspects from previous tasks.

First, we assume access to both initial ($\mathcal{D}_A$) and new datasets ($\mathcal{D}_B$), while typical CL based methods assume access only to the new data and to a model trained on the initial data. Xia et al. (2022) assumes access to a generative model trained on the initial dataset and trains the final model in a two-stage process. In contrast, we systematically derive PLM using a latent variable formulation and train our final model in a single stage.

Second, we do not constrain the new entity types to be disjoint from old entity types. For example, if PERSON is among the initial entity types, the new entity types can include ACTOR, which is a subtype of PERSON; our desired final model should predict a mention as (ACTOR, PERSON) if the mention is an actor, and just PERSON if the mention is a person but not an actor. This setup is related to prior work on hierarchical classification (Silla and Freitas, 2011; Arabie et al., 1996; Meng et al., 2019) or fine-grained NER (Ling and Weld, 2012; Choi et al., 2018; Mai et al., 2018; Ringland et al., 2019). However, the objective of such work is to use the hierarchical structure to train a better fine-grained NER model; they do not deal with taxonomy expansion. Abhishek et al. (2019) propose a unified hierarchical label set (UHLS) and use Partial Hierarchical Label Loss to train a unified model from multiple datasets with different (potentially overlapping) label sets. However, our work differs in two key aspects; first, our approach works even if two labels are overlapping but neither one is a subtype or supertype of other (i.e. even if the relation is non hierarchical) and second, our probabilistic approach is theoretically grounded (PLM).

Another class of related approaches is based on positive-unlabelled (PU) learning (Bekker and Davis, 2020; Grave, 2014; Peng et al., 2019) or learning from partial/noisy training data (Mayhew et al., 2019) — if we combine the initial ($\mathcal{D}_A$) and new datasets ($\mathcal{D}_B$), then the combined training data can be viewed as a positive-unlabelled or noisy dataset. However, unlike PU learning, the annotations in our setup are not randomly missing; instead, annotations of entity types $A$ and $B$ are missing in datasets $\mathcal{D}_B$ and $\mathcal{D}_A$ respectively.

## 3 Problem Definition and Notation

Given datasets $\mathcal{D}_A$ and $\mathcal{D}_B$ annotated with entity types $\mathcal{E}_A = \{e_A^1, e_A^2, ...e_A^m\}$ and $\mathcal{E}_B = \{e_B^1, e_B^2, ...e_B^n\}$ respectively, the goal of TE-NER is to learn a model to recognize entities from both $\mathcal{E}_A$ and $\mathcal{E}_B$. For our initial definition and solution, we assume that all entity types are distinct and mutually exclusive, i.e., a token does not belong to more than one entity type. While this is in line with prior work, it is also a shortcoming as entity types and definitions can often overlap in real world settings as well as academic datasets. For instance, in the FewNERD dataset (Ding et al., 2021), an entity can be both a PERSON as well as a POLITICIAN. To handle such cases, we define a more general version of the problem and solutions in §5.

Prior work on TE-NER is based on continual learning where the goal is to adapt the NER model continuously to new entity types (Monaikul et al., 2021; Xia et al., 2022). Such work assumes access to only the models trained on the original datasets ($Model_A$ trained on $\mathcal{D}_A$, $Model_B$ trained on $\mathcal{D}_B$) and not the datasets themselves. However, in many scenarios, both the original dataset and the new dataset are available but are annotated using different sets of entity types due to an evolving taxonomy or come from different sources. Hence, our definition assumes full access to both $\mathcal{D}_A$ and $\mathcal{D}_B$. Also, unlike few-shot and transfer-learning setups (Phang et al., 2018; Ma et al., 2022), our goal is to train a model ($Model_{final}$) that does well on both $\mathcal{E}_A$ and $\mathcal{E}_B$ (irrespective of the size of $\mathcal{D}_B$; however, we return to the question of size in §6.3.2).

**Why is Taxonomy Expansion challenging?** The central challenge with TE-NER is partial annotations — if a mention in $\mathcal{D}_A$ belongs to an entity in $\mathcal{E}_B$, it will not be annotated (e.g. in Fig 1, **BBC** is not annotated as ORGANIZATION). Similarly, $\mathcal{D}_B$ is also partially annotated. Such partial annotation misleads model training, and prior work attempts to mitigate this issue on a single dataset (Mayhew et al., 2019; Jie et al., 2019). We focus on this problem in the context of TE-NER.

## 4 Methods for TE-NER

We first discuss our methods for the scenario where all entity types are disjoint; then, in §5.3, we discuss modifications for a more general definition of TE-NER. We use the BIO scheme (Ramshaw and Marcus, 1995) for this work. All the approaches be-

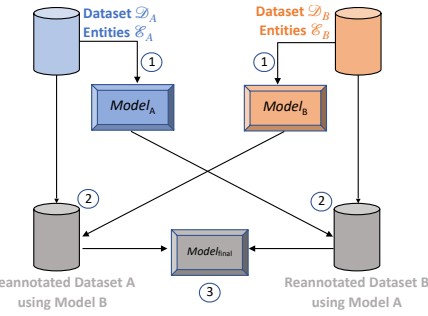

Figure 2: PLM consists of three steps: ① Train $Model_A$/$Model_B$ to recognize entities in $\mathcal{E}_A$/$\mathcal{E}_B$ using dataset $\mathcal{D}_A$/$\mathcal{D}_B$ ②Use $Model_A$ to annotate $\mathcal{D}_B$ with distribution over entity types $\mathcal{E}_A$ (repeat with $Model_B$ and $\mathcal{D}_A$ ③ Use these annotations with the PLM loss to train the final model. Cross-annotation is similar except in ②, we annotate the dataset with the hard predictions of the models instead of a distribution, and use a cross-entropy loss in ③. PLM-KL replaces the PLM loss in ③ with a KL-divergence term.

low are motivated by a simple observation: in both $\mathcal{D}_A$ and $\mathcal{D}_B$, the observed labels are not always the true labels. Specifically, if a word is annotated as O (used to indicate tokens that are not part of entity mentions), we do not know its true label. Figure 2 provides an overview of all approaches.

### 4.1 Cross Annotation (X-Ann.)

Before discussing our solution, we suggest a naive solution to TE-NER. As discussed before, the main challenge with TE-NER is partial annotations. If the annotations are correct and exhaustive, i.e., $\mathcal{D}_A$ was annotated for entity types $\mathcal{E}_B$ (and $\mathcal{D}_B$ for $\mathcal{E}_A$), then we could simply combine these datasets and train an NER model on the combined dataset. Cross-annotation is motivated by this observation — instead of expecting the data to be fully labeled, it uses model outputs to provide the missing labels.

Under our setup, if a token in $\mathcal{D}_A$ is annotated as O, it can still belong to either one of the entities in $\mathcal{E}_B$ (e.g. **BBC** in Figure 1 can truly be O or it can be an ORGANIZATION). Therefore, in $\mathcal{D}_A$, if a word is annotated as O, we replace it with the prediction of $Model_B$. Similarly, in $\mathcal{D}_B$, if a word is annotated as O, we replace it with the prediction from $Model_A$. We combine these re-annotated versions of $\mathcal{D}_A$ and $\mathcal{D}_B$ to train the final model.

### 4.2 Partial Label Model (PLM)

Cross annotation uses the hard labels obtained from the predictions of $Model_A$ and $Model_B$ to re-annotate $\mathcal{D}_A$ and $\mathcal{D}_B$. We present an alternate

approach that extends cross-annotation to use *soft* labels (i.e., the entire output distribution).

As in cross annotation, we start with the observation that the observed label for a token is not necessarily the true label, and conversely, the true label is unobserved. Thus, we treat the the true label as a latent variable. First, to simplify the discussion, we assume that the distribution of this latent variable is known and then solve for the optimal parameters of the model. Later, we relax this assumption and discuss how to approximate the distribution of this latent variable.

Let us denote our desired final model by $Model_{final}$, parameterized by $\theta$ as $f(s|\theta)$. The output of $f(\cdot)$ is a probability distribution over the entity labels $\mathcal{E}_{final} = \mathcal{E}_A \cup \mathcal{E}_B$, for each token in the input $s$ and the output for $i^{th}$ token is denoted by $f_i(\cdot)$. We use a BERT-based sequence tagging model for $f(\cdot)$ (§6.1).

First, let us calculate the likelihood for a single example $s \in \mathcal{D}_A \cup \mathcal{D}_B$ consisting of $n$ tokens, $s = [w_1, w_2, \ldots, w_n]$, and its corresponding observed labels $y = [y_1, y_2, \ldots, y_n]$. We denote the corresponding (latent) true labels as $Z = [Z_1, Z_2, \ldots, Z_n]$ and predicted labels as $Y = [Y_1, Y_2, \ldots, Y_n]$ with $Y_i \sim f_i(s|\theta)$ and $Z_i \sim g_i(s)$. Here $g()$ is an oracle function that gives us the distribution of true labels. The likelihood of the predictions matching the true underlying label, $P(Y = Z|\theta)$, is then calculated as

$$P(Y = Z|\theta_{final}) = \prod_{i=1}^{n} P(Y_i = Z_i).$$

Further, $P(Y_i = Z_i)$ can be decomposed as

$$P(Y_i = Z_i) = \sum_{e \in \mathcal{E}_{final}} f_i(s|\theta)[e] \times g_i(s)[e]$$
$$= \left\langle f_i(s|\theta), g_i(s) \right\rangle.$$

Finally, the negative log-likelihood yields the loss:

$$Loss(s|\theta) = \sum_{i=1}^{n} -\log \left\langle f_i(s|\theta), g_i(s) \right\rangle. \quad (1)$$

**Approximating $g(\cdot)$**   Equation 1 assumes access to an oracle function $g(\cdot)$ that gives us the distribution of the true labels. In practice, this is exactly the information that we do not have access to. Thus, we approximate $g$ using the predictions of a model, based on two key observations.

First, if a token $w_i \in s$ (for $s \in \mathcal{D}_A$) is annotated as an entity $e$, we know that the annotated label is the true label. In this case, $g_i(s)[e] = 1$, and $g_i(s)[e'] = 0 \ \forall e' \neq e$. Second, if $w_i$ is not annotated as an entity (i.e., is assigned an O label), then we know that it does not belong to any entity in $\mathcal{E}_A$. Therefore $g_i(s)[e] = 0 \ \forall e \in \mathcal{E}_A$. However, $w_i$ could potentially be an entity $\in \mathcal{E}_B$. So, we just need the probability distribution over $\mathcal{E}_B \cup \{O\}$,[1] which we can directly estimate by using $Model_B$.[2] Analogously, for $s \in \mathcal{D}_B$, we can use $Model_B$ to estimate the distribution over $\mathcal{E}_A$. With these approximations, Equation 1 can be split into two terms corresponding to the two cases above. For the first case, the loss is simply a cross-entropy loss against the one-hot vector obtained from $g_i(s)$. This gives us the proposed loss function of PLM.

$$Loss(s|\theta) = \sum_{i:y_i=O} CE\Big(f_i(s|\theta), \ y_i\Big)$$
$$+ \sum_{i:y_i \neq O} -\log \left\langle f_i(s|\theta), g_i(s) \right\rangle. \quad (2)$$

**PLM-KL:**   The loss term in Equation 2 is similar to the knowledge distillation loss, where the second term is replaced by a KL-divergence term:

$$Loss(s|\theta) = \sum_{i:y_i=O} CE\Big(f_i(s|\theta), \ y_i\Big)$$
$$+ \sum_{i:y_i \neq O} KL\Big(g_i(s) \ || \ f_i(s|\theta)\Big). \quad (3)$$

This loss term has been used in prior work on continuous learning in NER (Monaikul et al., 2021) and it simulates a student-teacher framework with $g$ as the teacher, and $f$ as the student. We prove, using a simple application of Jensen's inequality, that the loss in Equation 3 is an upper bound of the loss in Equation 2 (Appendix B). In §6, we experiment with both the exact loss function of PLM as well as the upper-bound of PLM-KL.

## 5   Revisiting Taxonomy Expansion

### 5.1   Definition

In §3, following prior work, we assumed that all the entity types are distinct and disjoint. We now

---

[1]More precisely, a token is not annotated with an entity type (e.g. PERSON), but a combination of B/I tag and the type (e.g. B-PERSON). We drop the B/I tags for simplicity, but the approach works identically regardless of the tags.

[2]$Model_B$ gives us the probability of not belonging to any of the $\mathcal{E}_B$, but since we already know that the token does not belong to $\mathcal{E}_A$, the probability of not belonging to $\mathcal{E}_B$ is equal to the probability of not belonging to $\mathcal{E}_A$ and $\mathcal{E}_B$.

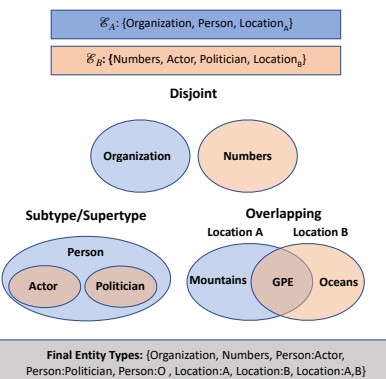

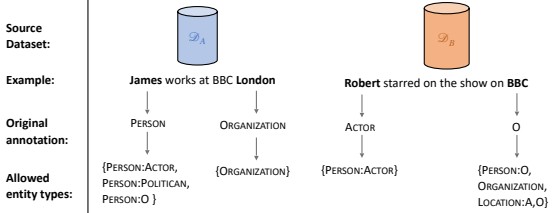

(a) Illustration of redefined output space

(b) Illustration of allowed entity types

Figure 3: For the non-disjoint case, the output space of the combined entity types is not simply a union of the two original entity types. 3a illustrates how the final entity types look like given the original types and the relations between them. 3b shows the allowed entity types in the redefined output space for each token in the dataset, given their original annotations.

extend our definition of the problem as well as methods to a more general version which allows for semantic overlap in the entity types. Specifically, we assume that along with $\mathcal{D}_A, \mathcal{D}_B, \mathcal{E}_A$, and $\mathcal{E}_B$, we also know the relationship $\mathcal{R}$, between the entity types ($\mathcal{E} = \mathcal{E}_A \cup \mathcal{E}_B$), where $\mathcal{R} : \mathcal{E} \times \mathcal{E} \rightarrow$ {DISJOINT, SUBTYPE, SUPERTYPE, OVERLAPPING}. The SUBTYPE/SUPERTYPE relations allow for an entity in $\mathcal{E}_B$ (e.g. Politician) to be a subtype of an entity in $\mathcal{E}_A$ (e.g. Person), and vice versa. The OVERLAPPING relation allows for partial overlaps in the definitions of types (e.g. if both $\mathcal{E}_A$ and $\mathcal{E}_B$ have a LOCATION type, but only a subset of LOCATIONS are common to both definitions).[3] Figure 3a illustrates these possible relationships.

**Output:** As in §3, our aim is to train a model to recognize both $\mathcal{E}_A$ and $\mathcal{E}_B$. However, now a mention can get more than one entity label. Consider

---

[3]We ignore the trivial case when an entity in $\mathcal{E}_A$ is exactly identical to $\mathcal{E}_B$. Without loss of generality, we also assume that entities within $\mathcal{E}_A$ ($\mathcal{E}_B$) are disjoint; even if they are not, we can always convert them to disjoint.

the SUBTYPE/SUPERTYPE example in Figure 3a, where a mention can belong to both {PERSON, ACTOR} or {PERSON, POLITICIAN}, or just PERSON (i.e. PERSON, but neither ACTOR nor POLITICIAN). Given this, we cannot directly train an NER model over $\mathcal{E}_A \cup \mathcal{E}_B$. Instead in $\mathcal{E}_{final}$, we define an entity type for each possible combination (illustrated in the grey box in Figure 3a), and train a model over the redefined output labels. For instance, we introduce three new entity types in the final label set corresponding to the PERSON entity for the three cases discussed above— PERSON:ACTOR, PERSON:POLITICIAN, and PERSON:O). Similarly, for the OVERLAPPING case, we define three new types — LOCATION:A (entities that are locations only according to $\mathcal{D}_A$), LOCATION:B (entities that are locations only according to $\mathcal{D}_B$), and LOCATION:A,B (entities that are locations according to both $\mathcal{D}_A$ and $\mathcal{D}_B$).

## 5.2 Allowed Entity Types

Before discussing the modifications to the methods for the non-disjoint case, we revisit the assumption that drove the methods in the disjoint case — we observed that if a token in $\mathcal{D}_A$ is annotated as O, it belongs to one of $\mathcal{E}_B \cup \{O\}$, else it belongs to the annotated entity type (§4.1). However, this assumption does not hold in the non-disjoint setting. In Figure 3b, **James** is annotated as PERSON in $\mathcal{D}_A$, so it can belong to one of {PERSON:ACTOR, PERSON:POLITICIAN, or PERSON:O}) in the final output space. Thus, for each token in the datasets, we define a set of *allowed entity types* that the token can belong to. These allowed types are determined by the observed annotation of that token and are a combination of existing entity types or the newly-introduced types as discussed above. Figure 3b gives more examples of allowed entity types.

## 5.3 Modifications to Proposed Methods

Given this mapping of the problem to detecting the entity types in a redefined output space, the modification to the methods from §6.3.1 lies in ② from Figure 2, where instead of annotating a token with predictions from a model, we simply constrain this annotation by the allowed entity types for that token. The rest of the steps proceed as before. We defer further details to Appendix C.

| Method | Ontonotes | FewNERD-Super | FewNERD-Sub | WNUT17 | JNLPBA | I2B2 |
|--------|-----------|---------------|-------------|--------|--------|------|
| Naive Join | 76.3 (2.1) | 68.4 (2.1) | 54.2 (1.6) | 27.9 (3.4) | 54.8 (3.5) | 76.5 (2.8) |
| CL | 87.7 (0.2) | 77.7 (0.1) | 66.1 (0.2) | 40.6 (1.3) | 70.5 (0.5) | 91.2 (0.3) |
| AML | 87.4 (0.3) | 77.9 (0.1) | 65.6 (0.2) | 38.4 (1.7) | 70.6 (0.4) | 91.6 (0.3) |
| X-Ann. | 87.8 (0.2) | 78.3 (0.1) | 66.9 (0.1) | 42.3 (1.2) | **71.5 (0.5)** | 92.0 (0.3) |
| PLM-KL | **88.2 (0.2)** | **78.4 (0.1)** | 67.0 (0.1) | **43.5 (1.1)** | 71.4 (0.5) | **92.5 (0.2)** |
| PLM | 88.1 (0.2) | **78.4 (0.1)** | **67.2 (0.1)** | 43.3 (1.3) | 71.4 (0.5) | 92.3 (0.3) |
| Upper Bound | 88.7 (0.2) | 78.6 (0.1) | 67.4 (0.1) | 45.3 (0.8) | 71.8 (0.4) | 93.3 (0.1) |

Table 1: Results for the disjoint setup (mean and std. dev. micro-F1 across 25 runs). PLM , PLM-KL, and cross annotation are competitive across the board and close to the upper bound.

## 6 Experiments

### 6.1 Datasets, Setup, and Hyperparameters

We study TE-NER using datasets covering diverse domains, entity types, and sizes—**(1)** Ontonotes (Weischedel et al., 2013) **(2)** WNUT17 (Derczynski et al., 2017) **(3)** JNLPBA (Kim et al., 2004) and **(4)** I2B2 (Stubbs and Uzuner, 2015). Since these datasets are fully annotated, we cannot use them directly to study TE-NER. Instead we modify each dataset to obtain partial annotations. For the non-disjoint setup, we only use the FewNERD dataset. We defer more details to Appendix A.

For all experiments, we finetune BERT-base as our backbone models for all experiments with the exact setup from Devlin et al. (2019). We repeat every experiment with 5 random splits of $\mathcal{D}_A$ and $\mathcal{D}_B$ and 5 different seeds (for training) for each split and report mean and standard deviation of micro-F1 scores averaged across all $5 \times 5$ runs. For each dataset, we use the validation data to choose the best learning rate from $\{5e^{-6}, 1e^{-5}, 2e^{-5}, 3e^{-5}, 5e^{-5}\}$.

### 6.2 Baselines and Upperbound

**Naive Join:** A naive solution to TE-NER is to combine the two partially annotated datasets $\mathcal{D}_A$ and $\mathcal{D}_B$ and train a model on this combined dataset using a cross-entropy loss. We expect this approach to perform poorly, but it highlights the severity of issues caused by partial annotation.

**Continual Learning (CL):** This baseline uses knowledge distillation similar to Monaikul et al. (2021) and Xia et al. (2022). Briefly, we first train $Model_A$ as teacher, then we train the student model ($Model_{final}$) on $\mathcal{D}_B$; if a token is annotated as $e \in \mathcal{E}_B$, we use the cross-entropy loss, else we calculate KL divergence with respect to the teacher's output.

**Modified Continual Learning (CL++):** Default CL does not work when the entity types are not disjoint. CL++ is a modified version of CL for our non-disjoint setup to account for allowed entity types.

**Adjusted Multilabel (AML):** The AML baseline treats the NER problem as a multi-label classification (with sigmoid loss) problem for each token, instead of multi-class as in a standard sequence tagging approach. However, instead of all entities, the loss for multi-label considers the loss for only the allowed entity types.

**Upperbound:** Since $\mathcal{D}_A$ and $\mathcal{D}_B$ are derived from fully annotated datasets $\mathcal{D}$, where every example is annotated for $\mathcal{E}_{final}$, we train our model on the original, unmodified dataset $\mathcal{D}$ and use it as a hypothetical upperbound. In real scenarios, we do not have such fully annotated datasets.

### 6.3 Research Questions and Experiments

We focus on three key research questions:

1. Is PLM more accurate than other approaches given *sizeable* training data ($\mathcal{D}_B$) for the new entity types $\mathcal{E}_B$?
2. Is PLM more accurate than other approaches given *little* training data ($\mathcal{D}_B$)?
3. Does PLM show robust performance when validation data is not exhaustively annotated?

#### 6.3.1 Accuracy of PLM given *sizeable* $\mathcal{D}_B$

We answer our first research question via experiments that focus on three different scenarios for TE-NER — DISJOINT (§3), SUBTYPES/SUPERTYPE, and OVERLAPPING (§5.1).

**Disjoint Entity types** Table 1 shows the results of all methods on the disjoint setup. First, Naive Join performs significantly worse than all other methods, with a drop of as much as 16 F1 in the case of JNLPBA. This is an expected outcome,

| Method | PER. | LOC. | ORG. | PROD. | PER. | LOC. | ORG. | PROD. |
|---|---|---|---|---|---|---|---|---|
| CL++ | 73.4 (0.2) | 76.2 (0.1) | 75.6 (0.1) | 77.0 (0.1) | 74.7 (0.1) | 75.6 (0.1) | 75.1 (0.1) | 76.8 (0.1) |
| AML | 72.7 (0.2) | 75.2 (0.1) | 74.5 (0.1) | 76.5 (0.1) | 74.8 (0.1) | 75.6 (0.1) | 75.1 (0.1) | 76.8 (0.1) |
| X-Ann. | 74.1 (0.1) | 76.9 (0.1) | 76.3 (0.1) | 77.7 (0.1) | 75.5 (0.1) | **76.4 (0.1)** | 75.9 (0.1) | **77.6 (0.1)** |
| PLM-KL | 74.1 (0.1) | 76.8 (0.1) | 76.2 (0.1) | 77.7 (0.1) | 75.5 (0.1) | 76.3 (0.1) | 75.8 (0.1) | 77.5 (0.1) |
| PLM | **74.2 (0.1)** | **77.0 (0.1)** | **76.4 (0.1)** | **77.8 (0.1)** | **75.6 (0.2)** | **76.4 (0.1)** | **76.0 (0.1)** | **77.6 (0.1)** |
| Upper Bound | 74.6 (0.1) | 77.1 (0.1) | 76.6 (0.1) | 78.0 (0.1) | 76.1 (0.1) | 76.9 (0.1) | 76.6 (0.1) | 77.9 (0.1) |

Table 2: Results for the SUBTYPE/SUPERTYPE (left side), and OVERLAPPING (right side) setups (mean and std. dev. micro-F1 across 25 runs). Each column indicates the entity type manipulated to create datasets (§6.1).

and it highlights the severity of problem caused by partial annotation. Second, both CL and AML are more accurate than the Naive Join approach as they approach the upper bound. Next, despite its simplicity, cross annotation offers a very strong solution to this problem. For 4 datasets, cross annotation is within 1 F1 of the Upper Bound and CL and AML are both behind cross annotation.

Finally, both PLM and PLM-KL reach scores that are on par, or even slightly better than cross annotation, indicating that they offer alternative solutions to this problem. However, the differences of these methods with cross annotation are very small, and we take a closer look in future sections.

**Non-disjont Entity Types:** We now turn to the non-disjoint setup (§5), using the FewNERD dataset (§6.1). Table 2 shows the results of these experiments, with the left-side showing the results for the SUBTYPES/SUPERTYPES case, and the right-side showing the results for the OVERLAPPING case. In either case, each column indicates the entity type manipulated to create datasets per Appendix A. We leave out the Naive Join approach given its (expectedly) poor performance in the previous experiment.

The observations for this set of results are very similar to the previous experiments and also consistent across the SUBTYPES/SUPERTYPES and OVERLAPPING — cross annotation again proves to be a highly effective solution, both PLM and PLM-KL are also on par with cross annotation, and all three methods approach the Upper Bound.

### 6.3.2 Accuracy of PLM given *small* $\mathcal{D}_B$

A reasonable assumption in TE-NER is that the dataset with the new entities ($\mathcal{D}_B$) will be much smaller than the initial dataset ($\mathcal{D}_A$) since it is impractical to annotate as many examples for each new entity type as exist for the old entity types. Our second set of experiments aims to test how various methods perform in scenarios where the number

of examples in $\mathcal{D}_B$ is limited, while maintaining the size of $\mathcal{D}_A$. We only focus on cross annotation and PLM as these were most competitive in the experiments in §6.3.1. Further, we focus only on the Ontonotes and FewNERD datasets.

Table 3 indicates that in few-shot settings, PLM consistently outperforms cross annotation across the board. The gap between the two methods is more severe as the number of examples reduce, with PLM scoring as much as 10 F1 higher when $\mathcal{D}_B$ contains only 100 examples for each type on Ontonotes, and 17 F1 higher on FewNERD with 300 examples per type. However, both approaches tend to be very unstable in such low data regimes, indicated by the high variance in results (more severe in Ontonotes). Despite this, PLM exploits the soft labels from the models to achieve better scores.

### 6.3.3 Robustness to partially annotated validation data

Experiments so far have assumed access to a fully annotated validation set. In practice, it is likely that even validation sets are partially annotated (similar to training sets). How do models behave with such partially annotated validation sets? Specifically, we assume that we have validation sets, $\mathcal{D}_A^{val}$ and $\mathcal{D}_B^{val}$ corresponding to $\mathcal{D}_A$ and $\mathcal{D}_B$ respectively. During the validation step, we evaluate the model being trained ($Model_{final}$) on both $\mathcal{D}_A^{val}$ and $\mathcal{D}_B^{val}$ separately, masking out any predictions from $\mathcal{E}_B$ on $\mathcal{D}_A^{val}$ and from $\mathcal{E}_A$ on $\mathcal{D}_B^{val}$. We use the average micro-F1 on these individual validation sets as our stopping criterion. Again, we focus on cross annotation and PLM on Ontonotes and FewNERD.

Table 4 shows that F1 scores with partial validation are similar to those of full validation. Thus, we do not even need validation data labeled with entities from both $\mathcal{E}_A$ and $\mathcal{E}_B$, as partially annotated validation sets are a reasonable proxy.

| Dataset | Method | 100 | 200 | 300 | 500 | 1000 | 2000 |
|---------|--------|-----|-----|-----|-----|------|------|
| OntoNotes | X-Ann. | 24.8 (16.7) | 6.2 (3.1) | 6.0 (3.1) | 57.9 (6.1) | 71.0 (1.4) | 80.1 (1.0) |
|  | PLM | 37.0 (24.6) | 16.5 (11.8) | 16.5 (11.6) | 60.5 (5.5) | 75.8 (1.4) | 81.2 (0.9) |
| FewNERD | X-Ann. | - | 14.4 (10.0) | 8.2 (4.1) | 42.6 (7.7) | 48.1 (4.3) | 55.9 (1.4) |
|  | PLM | - | 28.2 (4.0) | 25.5 (5.5) | 47.4 (7.8) | 54.2 (2.3) | 60.1 (0.5) |

Table 3: Experiments with varying sizes of $\mathcal{D}_B$ (mean and std. dev. micro-F1 across 25 runs). Each column indicates the number of examples per type. As $|\mathcal{D}_B|$ reduces, PLM scores higher F1 compared to cross annotation.

| Dataset | Val. type | X-Ann. | PLM |
|---------|-----------|--------|-----|
| OntoNotes | Full val. | 87.7 (0.2) | 88.1 (0.2) |
|  | Partial val. | 87.7 (0.2) | 88.1 (0.2) |
| FewNERD | Full val. | 66.8 (0.2) | 67.1 (0.2) |
|  | Partial val. | 66.8 (0.2) | 66.7 (0.2) |

Table 4: F1 scores of PLM and cross-annotation are almost identical regardless of whether validation data are partially or fully annotated.

| $|\mathcal{E}_A|$: $|\mathcal{E}_B|$ | X-Ann. | PLM |
|---------|--------|-----|
| 9:9 | 87.8 (0.2) | 88.1 (0.3) |
| 12:6 | 87.7 (0.2) | 88.1 (0.2) |
| 15:3 | 87.7 (0.2) | 88.1 (0.2) |
| 16:2 | 87.7 (0.2) | 88.0 (0.2) |
| 17:1 | 87.9 (0.3) | 88.1 (0.2) |

Table 5: Results with varying $|\mathcal{E}_A|$ and $|\mathcal{E}_B|$; F1 scores of cross annotation and PLM do not vary with changes in $|\mathcal{E}_A|$ and $|\mathcal{E}_B|$.

### 6.3.4 Summary of results

Overall, cross annotation and PLM are similarly accurate when $\mathcal{D}_B$ is large enough. Both methods are also robust when partially annotated validation sets are used instead of fully annotated validation sets. However, as the size of $\mathcal{D}_B$ reduces, PLM is increasingly more accurate.

## 7 Discussion and Further Analysis

Having seen the effectiveness of cross annotation and PLM, we further analyze these methods. For these experiments, we use Ontonotes under the disjoint setting (§6.3.1).

### 7.1 Effect of size of $\mathcal{E}_A$ and $\mathcal{E}_B$

In experiments in §6, $\mathcal{E}_A$ and $\mathcal{E}_B$ were set to have (almost) the same number of entity types. However, a more likely scenario is that the number of entity types to be added ($|\mathcal{E}_B|$) are fewer than the number of existing entity types ($|\mathcal{E}_A|$). We investigate the behavior of methods in such settings — for these experiments we keep the size of the datasets ($\mathcal{D}_A$ and $\mathcal{D}_B$) to be fixed and identical to those in

| $|\mathcal{E}_A|$: $|\mathcal{E}_B|$ | $Model_B$ | XAnn. | PLM |
|---------|-----------|-------|-----|
| 9:9 | 87.7 (2.4) | 88.2 (2.1) | 88.5 (2.1) |
| 12:6 | 87.3 (3.1) | 87.8 (3.0) | 88.3 (2.8) |
| 15:3 | 84.8 (8.1) | 85.5 (8.3) | 86.3 (7.1) |
| 16:2 | 86.0 (8.9) | 87.3 (7.2) | 87.8 (6.4) |
| 17:1 | 72.1 (13.3) | 74.2 (12.4) | 75.7 (11.9) |

Table 6: The model from PLM recognizes $\mathcal{E}_B$ more accurately than $Model_B$. As $|\mathcal{E}_A|$ increases and $|\mathcal{E}_B|$ decreases, this performance gap increases.

§6. Results in Table 5 indicate that even as the difference in number of entity types in $\mathcal{E}_A$ and $\mathcal{E}_B$ grows, cross Annotation and PLM do not diverge in behavior and continue to be equally accurate.

### 7.2 Performance of $Model_{final}$ Vs $Model_B$

Another possible solution to TE-NER is to use $Model_B$ to recognize the new entity types from $\mathcal{E}_B$, $Model_A$ to recognize the original entity types $\mathcal{E}_A$, and combine the predictions of these two models via heuristics. However, this begs the question — does $Model_B$ recognizes entities from $\mathcal{E}_B$ as well as (or better than) $Model_{final}$ derived from Cross Annotation or PLM? We answer this by testing these methods against $Model_B$ on a test set annotated with just $\mathcal{E}_B$. If $Model_{final}$ predicts a mention as $e \in \mathcal{E}_A$, we simply ignore it.

Results (Table 6) indicate that **(1)** PLM and cross annotation are consistently more accurate than $Model_B$ at recognizing entity types from $\mathcal{E}_B$, and **(2)** as $|\mathcal{E}_B|$ decreases (and $|\mathcal{E}_A|$ increases), the gap between all methods increases. We hypothesise that this is due to three reasons: (1) Cross Annotation and PLM use additional data corresponding to $\mathcal{E}_A$ in dataset $\mathcal{D}_A$ hence their superior performance. (2) The more such additional information (about other entity types) present in $\mathcal{D}_A$, the larger the performance gain. Intuitively, this implies that the ability to recognize entity types in $\mathcal{E}_A$ is helping in better recognizing entities of interest in $\mathcal{E}_B$.

## 8 Conclusion

We define and propose solutions for a general version of the problem of taxonomy expansion for NER. Crucially, and unlike prior work, our definition does not assume that the entity types that are being added are disjoint from existing types. We propose a novel approach, PLM, that is theoretically motivated based on a latent variable formulation of the problem, and show that prior solutions based on student-teacher settings are approximations of the loss arrived at by our method. PLM outperforms baselines on various datasets, especially in data scarce scenarios when there is limited data available for the new entity types. In such settings, it is as much as 10-15 F1 points more accurate than the next best baseline.

## 9 Limitations

There are many other extensions of the definition and setup for TE-NER that this work does not address. For instance, the old and the new entity types / datasets, can belong to different domains and results in such settings are likely to different than those reported in this paper, where both old and new entity types are created from the same pre-annotated dataset. Studying this, however, requires creation of appropriate datasets, which is also something that this work does not attempt to do.

For the more general definition of TE-NER that allows entity types to be related, We assume that we know these relations (i.e. they have been provided by an expert/user) and leave aside the problem of how to identify these relations in the first place. A separate body of work in taxonomy induction aims to identify such a hierarchy between entities (e.g. (Snow et al., 2006)).

Finally, despite its stronger theoretical foundations, the key method proposed in this work, PLM, is not more accurate than the simple cross annotation baseline in data rich scenarios, and is more suitable for data scarce scenarios. Further investigation is required to boost results for the former scenario.

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

## A   Dataset creation process

We create partially annotated datasets from a fully annotated dataset $\mathcal{D}$ with entity types $\mathcal{E}$ by first splitting the $\mathcal{E}$ into two (approximately) equal subsets (yielding $\mathcal{E}_A$ and $\mathcal{E}_B$), and then repeating this with the examples in $\mathcal{D}$ (yielding $\mathcal{D}_A$ and $\mathcal{D}_B$). Additionally, for the examples, we also scrub the annotations corresponding to the complementary entity types (e.g., remove entity annotations corresponding to $\mathcal{E}_A$ from $\mathcal{D}_B$).

For validation and test data, we use the corresponding fully annotated development and test sets as it is respectively. However, the assumption of fully annotated validation data being available is unrealistic, and we return to this in §6.3.3

**Datasets for Non-disjoint TE-NER**  We study the non-disjoint version of the problem (§5.1) using the FewNERD dataset (Ding et al., 2021). This dataset consists of two levels of annotations corresponding to coarse-grained supertypes (e.g., PERSON) and fine grained subtypes (e.g., ACTOR).[4]

To study the SUBTYPE/SUPERTYPE scenario, after we split the coarse-grained entity types $\mathcal{E}$ into $\mathcal{E}_A$ and $\mathcal{E}_B$ as above, we randomly add a subset of the subtypes corresponding to a coarse-grained entity type $e \in \mathcal{E}_A$ to $\mathcal{E}_B$. For instance, if the PERSON entity type is present in $\mathcal{E}_A$, we add a subset of its eight subtypes (POLITICIAN, ACTOR, ARTIST etc.) to $\mathcal{E}_B$.

To study the OVERLAPPING scenario, we first choose a coarse-grained entity type (e.g. PERSON), split its fine-grained subtypes into two overlapping subsets $S_A$ and $S_B$ (e.g. $S_A$ = ACTOR, POLITICIAN, ..., and $S_B$ = ACTOR, ARTIST, ...). Then, if a mention in $\mathcal{D}_A$ is annotated with a subtype in $S_A$, we assign it a new entity type PERSON$_A$. Similarly, if a mention in $\mathcal{D}_B$ is annotated with a subtype in $S_B$, we assign it a new entity type PERSON$_B$. PERSON$_B$ and PERSON$_B$ are added to $\mathcal{E}_A$ and $\mathcal{E}_B$ respectively.

## B  Proof of lower bound

To show that the loss term for PLM in Equation 2 is a lower bound on the KL-divergence loss used in prior work (Equation 3), we start with the loss in Equation 2.

$$Loss(s|\theta) = \sum_{i:y_i=O} CE\Big(f_i(s|\theta),\, y_i\Big)$$
$$+ \sum_{i:y_i\neq O} -\log\Big\langle f_i(s|\theta),\, g_i(s)\Big\rangle. \quad (4)$$

[4]For the disjoint version of the problem (§6.3.1), we simply treat these two different levels of the dataset as two different datasets — FewNERD-Super and FewNERD-Sub.

Using Jensen's inequality for concave functions (like $\log x$), we get $\log E[x] \geq E[\log x]$. Hence,

$$Loss(s|\theta) \leq \sum_{i:y_i=O} CE\Big(f_i(s|\theta),\, y_i\Big)$$
$$+ \sum_{i:y_i\neq O}\Big\langle -\log(f_i(s|\theta)),\, g_i(s)\Big\rangle.$$
$$\leq \sum_{i:y_i=O} CE\Big(f_i(s|\theta),\, y_i\Big)$$
$$+ \sum_{i:y_i\neq O}\Big\langle -\log(\frac{f_i(s|\theta)}{g_i(s)}),\, g_i(s)\Big\rangle.$$
$$= \sum_{i:y_i=O} CE\Big(f_i(s|\theta),\, y_i\Big)$$
$$+ \sum_{i:y_i\neq O} KL\Big(g_i(s)\,||\,f_i(s|\theta)\Big). \quad (5)$$

## C  Modifications to proposed methods

### C.1  Modified Cross Annotation

Recall that the idea behind cross-annotation is to use model predictions as a proxy for actual annotations. For the non-disjoint case, if a token in $\mathcal{D}_A$ is annotated as $e$, then we first get its allowed entity types following §5.2. If this set of allowed entities contains just one element, then we annotate it as the allowed entity type. Otherwise, we use $Model_B$ to chose the best label among the allowed entities. For the example in Figure 3b, **James** is annotated as PERSON in $\mathcal{D}_A$, so we know that its allowed entity types are {PERSON:ACTOR, PERSON:POLITICIAN, PERSON:O }. We use $Model_B$ to get the probability of ACTOR, POLITICIAN and $O$ and among them, choose the one with highest probability. After re-annotating both $\mathcal{D}_A$ and $\mathcal{D}_B$, the rest of the steps proceed as in §4.1.

### C.2  Modified PLM

For PLM, if we are given the distribution of true labels ($g_i(s)$), the likelihood calculation does not change, therefore the loss in Equation 1 remains the same for the non-disjoint case. However, similar to cross annotation above, the estimation of the oracle function $g(.)$ changes. For a data point in $\mathcal{D}_A$ (similarly for $\mathcal{D}_B$), if a token is annotated as $e$, we need $g(.)$ to give a probability distribution over its allowed entity types (the probability of other entity types is 0). Therefore, we compute a softmax over the logits of the allowed entity types. In Figure 3b, **James** is annotated as PERSON, so we use $Model_B$ to get the logits of ACTOR, POLITICIAN,

and O, and then take softmax over just these (the
probability of other entities in $\mathcal{E}_{final}$ is 0)