# OpenReview forum: "Taxonomy Expansion for Named Entity Recognition"
_EMNLP/2023/Conference — EMNLP 2023 Main_

### Official Review · Reviewer_451n · 2023-07-20

**Soundness:** 4

**Excitement:**

4: Strong: This paper deepens the understanding of some phenomenon or lowers the barriers to an existing research direction.

**Paper Topic And Main Contributions:**

The paper addresses the problem of taxonomy expansion for NER, i.e. increasing the set of entity labels that the NER model should recognize, under the assumption that re-annotation of the original dataset is impractical, and instead a different dataset with annotations for the novel entity labels is available. To address this problem of training on two disjoint, partially labeled datasets, the paper proposes a variant of cross-annotation that uses the output distribution as soft labels and a latent variable formulation of the loss. The authors also introduce a novel variant that allows for semantic overlap of the entity sets (e.g. sub/supertypes). Experiments on 6 datasets show that their approach is on-par with/outperforms standard cross-annotation, and is especially beneficial in low-resource settings with few examples per entity type. (main contribution are NLP engineering experiment / approach for data efficiency).

**Questions For The Authors:**

- Question A: Are the results reported token- or entity-level micro F1?

- Question B: This is more a suggestion, but there was no entry field for suggestions, so I put it here: Due to the way the dataset splits are constructed from a single underlying fully annotated dataset, the results reported in this paper present an optimal case where the underlying distribution of entity labels is the same for the two datasets DA and DB. As the authors acknowledge in the introduction, this is typically not the case, as the "old" dataset DA is often sampled with respect to the original entity type set, and the new dataset DB is (most likely) focused on the new entity type set, even if it is drawn from the same general 'domain', e.g. news text. It would have been interesting the see results for this scenario as well, e.g. by using ConLL-03 (news) and Ontonotes (news + dialogue). Did you do any experiments in this direction?



**Reasons To Accept:**

- Interesting problem and a real-world issue, with the proliferation of “disjoint” datasets that often need to be combined
- paper is well written and easy to follow
- interesting extension to 'hard' cross-annotation
- experiments are extensive, and results are convincing

**Reasons To Reject:**

- I can't find any

**Reproducibility:**

4: Could mostly reproduce the results, but there may be some variation because of sample variance or minor variations in their interpretation of the protocol or method.

**Reviewer Confidence:**

3: Pretty sure, but there's a chance I missed something. Although I have a good feel for this area in general, I did not carefully check the paper's details, e.g., the math, experimental design, or novelty.

**Typos Grammar Style And Presentation Improvements:**

- line 13, whitespace between PLM and consistency
- line 21, ‘a more cost effective approach ‘
- line 463, ‘Non-disjoint’
- line 606, ‘we’
- line 786. end-of-sentence period

---

> ### Author Rebuttal · Authors · 2023-08-28
>
> Thank you for your review. All typos/presentation comments will be addressed in the next version of the paper.
>
> Answers to questions:
> 1. We use default “seqeval” evaluation metrics so these would be entity-level metrics. We will clarify this in the next version of the paper.
> 2. Yes, that is an interesting experiment to analyze and we considered it but left it out of scope (see Limitations section). However, one of the major challenges is the test set. Say we have Domain A data annotated with entity types A and Domain B data annotated with entity type B. To test our approach we need to reliable test set set in Domain B annotated with both entity types A and B; such data is relatively harder to find. A possible workaround could be to remove some of the common entity types like PER, LOC from a dataset and this forms the Domain B and use a dataset from a different domain for those common entity types. However, there could still be differences in annotation guidelines across different datasets, so the validity of findings would be slightly questionable.

---

### Official Review · Reviewer_CUSJ · 2023-07-31

**Soundness:** 4

**Excitement:**

4: Strong: This paper deepens the understanding of some phenomenon or lowers the barriers to an existing research direction.

**Missing References:**



**Paper Topic And Main Contributions:**

This paper is about extending named entity annotation with newer categories, called "taxonomy expansion". It proposes a method that composes two annotated resources to provide a better system for annotation with the merge of the categories. It provides extensive comparuison with other methods, as well as several empirical tests for the methods it suggests, on the size of the second corpus and on partial annotation.

**Questions For The Authors:**

Question A: Your examples are always (probably realistically) on the way to more specification of classes. Have you considered the possibility of ambiguating? That is, having writers, actors and directors in an initial annotation, and people in a second corpus, could you convert everything just to people?

**Reasons To Accept:**

The issue is very well explained, the methods are robust and provide good results.

**Reasons To Reject:**

I can see none.

**Reproducibility:**

5: Could easily reproduce the results.

**Reviewer Confidence:**

4: Quite sure. I tried to check the important points carefully. It's unlikely, though conceivable, that I missed something that should affect my ratings.

---

> ### Author Rebuttal · Authors · 2023-08-28
>
> Thank you for your review.
>
> In the final predictions, if the model predicts “actor”, we can still deterministically assign this entity as “person”, i.e, by predicting “actor” we do not lose any information since “actor” is the more specific tag. In general, if annotations go from specific to generic, we can always do such a deterministic mapping.

---

### Official Review · Reviewer_nj9h · 2023-08-08

**Soundness:** 2

**Excitement:**

3: Ambivalent: It has merits (e.g., it reports state-of-the-art results, the idea is nice), but there are key weaknesses (e.g., it describes incremental work), and it can significantly benefit from another round of revision. However, I won't object to accepting it if my co-reviewers champion it.

**Missing References:**

Authors may want to check with the following related work, where entity type expansion is considered with the assumption that complex relations among types may exist.

Abhishek, A P Azad, B Ganesan, A Anand, A Awekar. Collective Learning From Diverse Datasets for Entity Typing in the Wild. 2nd International Workshop on Entity Retrieval (EYRE), CIKM 2019. [https://ceur-ws.org/Vol-2446/paper3.pdf]

**Paper Topic And Main Contributions:**

Several tasks in NLP and other areas often face the challenges of handling a greater number of classes than any single labeled dataset provides. This paper focuses on the problem of taxonomy extension or increase in the number of classes for Named Entity Recognition (NER) task. Toward this objective, it proposes models for two distinct scenarios:
1.	Entity types coming from different datasets are disjoint. For this scenario, they suggested two solutions. The first solution is based on cross-annotation (X-Ann.), where models trained on one dataset is used to get annotation of additional entity types for the other dataset and vice-versa. Then the combined set of re-annotated datasets is used to train the final model. X-Ann. approach is quite trivial and intuitive. The second solution (Partial Label Model or PLM) treats the prediction of additional entity types as soft-label. In the PLM framework, a modified loss function resembling the student-teacher framework is used.
2.	Entity types in different datasets may be related. In this scenario, entity types present in different datasets are assumed to be related. In particular, four relations between a pair of entity types were considered: disjoint, subtype, supertype and overlapping. Based on these relations, entities are relabeled with a modified set of final entity types. The final set of entity types, however, is made flat, even if one of the datasets has hierarchical labels.

The experiments are performed on six datasets covering diverse domains and entity types.


**Questions For The Authors:**

1. Why are the experiments performed separately for subtype/supertype and overlapping relations among entity types?
2. Why the choice of baseline methods was kept limited?


**Reasons To Accept:**

1. Relevant problem


**Reasons To Reject:**

1.	Limited set of baseline methods was selected
2.	Experiment setup is basic and not matching with the ideal situation. Why the experiments are performed separately for subtype/supertype and overlapping relations among entity types?
3.	Limited set of entity types is considered. Perhaps, that is the reason, even the basic X-Annot. method has similar performance as with other methods.
4. Manual intervention to get the final sets of entity types if relations among them exist. It may not be a good idea when entity types increase from 100s to 1000s.


**Reproducibility:**

4: Could mostly reproduce the results, but there may be some variation because of sample variance or minor variations in their interpretation of the protocol or method.

**Reviewer Confidence:**

4: Quite sure. I tried to check the important points carefully. It's unlikely, though conceivable, that I missed something that should affect my ratings.

---

> ### Author Rebuttal · Authors · 2023-08-28
>
> Thank you for your review and feedback.
>
> * “Limited baseline”: Please see the response to question 2 below.
>
> * “manual intervention”:  In cases of thousands of labels, we can probably use LLM if the label names are meaningful, or even without any external knowledge, we can probably estimate the relation based on the model’s predictions and gold annotations. Regardless, the effort to identify such relation is still significantly smaller than actually annotating the data for all the entity types, and reducing such effort is one of the goals of the paper.
>
> * “Limited set of entity types”: We wonder why the reviewer thinks the set of entity types is limited. FewNerd consists of 8 coarse-grained entity types, and 66 fine-grained types. Ontonotes consists of 18 entity types, while i2b2 also consists of 18 entity types spanning PII entities in the medical domain. In fact, we consciously included multiple datasets from different domains to ensure that out conclusions are broadly applicable, so we would like more clarity on the specific limitations.
>
> * “Related Work” : Thank you for pointing us to a related paper, we would be happy to cite and discuss their approach in related works. However, we would like to point out that there are also  significant differences with that work. In particular, we propose a theoretically grounded probabilistic approach, whereas their approach is similar to CL++  (not exactly same but comparable; snippet from the reference “From this normalized distribution, we select a label which has the highest probability and is also a member of the mapped labels Ym. We assumed the selected label to be correct and propagate the log-likelihood loss.” )
>
> Below are the answers to your questions.
>
> 1. We consider subtype/supertype and overlapping separately because it is a relatively controlled set of experiments with results that are easier to analyze. If we combine both cases, then in each experiment we have two categories like PER for subtype and LOC for overlapping. In addition, there is also randomness in how we create overlapping categories in LOC. Visualizing all of them along with different solutions could become quite challenging. Nonetheless, our solution is applicable in combination too, and we expect it to work as well as it does in individual cases.
>
> 2. The exact problem definition of our work is novel, even though it is similar to problems in continual learning space (which we compared), it's not exactly the same. Moreover, the seemingly simple baselines we choose, indeed performs very well on the task on a large data scenario. If you can further point out in what ways the baselines are limited, we would be happy to discuss this in future iterations of the paper.

---

### Official Review · Reviewer_xZ9p · 2023-08-10

**Soundness:** 4

**Excitement:**

3: Ambivalent: It has merits (e.g., it reports state-of-the-art results, the idea is nice), but there are key weaknesses (e.g., it describes incremental work), and it can significantly benefit from another round of revision. However, I won't object to accepting it if my co-reviewers champion it.

**Paper Topic And Main Contributions:**


This paper presents a comprehensive study on the taxonomy expansion problem for named entity recognition. The authors formally define the expansion problem allowing overlap between old and new entity types, and propose the Partial Label Model (PLM) algorithm based on partially labeled modeling. Experiments show that PLM can improve over baselines in low-resource scenarios, and achieve competitive results even with complex subtype and overlapping relations between new and old types. Analysis demonstrates models trained with PLM recognize new types more accurately than those trained only on the new type data, especially when the number of new types is small.  In summary, the paper provides a formal definition for the NER taxonomy expansion problem, and proposes a theoretically grounded PLM solution that shows significant advantages in low-resource settings.

**Questions For The Authors:**

1. Does your method also consider the dynamic scenario where the number of categories continuously increases? Does the model experience the problem of forgetting during the continuous learning process?
2. In the experiments, is the sample category distribution kept balanced? How stable and robust is the model when the categories are imbalanced?
3.  If the performance of PLM is similar to simple cross-annotation, then where lies the advantage of PLM? Are there any comparison experiments in complex scenarios?

**Reasons To Accept:**

1. A Partial Label Model (PLM) method based on latent variable modeling has been proposed. PLM uses individually trained models to provide soft labels for another dataset, and then trains the final model. This method is quite reasonable.

2.  Compared to other methods, it is shown that PLM performs well when there is limited data for new categories, and PLM is also suitable for situations where categories overlap.

**Reasons To Reject:**

1. The paper assumes that the relationship between new and old categories (non-overlapping, subclasses, etc.) is known. However, in reality, a category relationship judgment module is also needed.
2. The theoretically more optimized PLM algorithm is similar in performance to simple cross-annotation and does not show a significant advantage in large data scenarios.

**Reproducibility:**

3: Could reproduce the results with some difficulty. The settings of parameters are underspecified or subjectively determined; the training/evaluation data are not widely available.

**Reviewer Confidence:**

4: Quite sure. I tried to check the important points carefully. It's unlikely, though conceivable, that I missed something that should affect my ratings.

---

> ### Author Rebuttal · Authors · 2023-08-28
>
> Thank you for your valuable feedback.
>
> * “category relationship judgment module” : We would argue that it is not necessary to model this problem. This can be a simple input from users. Identifying such relationships is a far simpler task than fully annotating all datasets for all types.
>
>
>
> Answers for the questions:
> 1. A simple solution to the dynamic scenario is to train the models incrementally – first train a model using PLM to recognize entity A and B, call it Model-AB, then train a model C to recognize entity type C, now again use PLM to train the model-ABC. In our experiments, this s the model’s probabilistic annotations are similar to gold annotations. In data scarce scenarios, this may not work, because of error propagation; instead we can train Model-AB, Model-BC and Model-CA and use then use them to (probablistically) annotate Dataset C, A and B respectively and then train the final model-ABC. However, the caveat with this approach is scalability. We would be happy to discuss and include preliminary results in the appendix of the next version of the paper, with both these approaches.
> 2. In Table 5 / Section 7.1, we analyze the effect of difference in the number of entity types in set A and B, which naturally leads to an imbalance in the number of mention in the two sets. The results are very robust and an imbalance in the distribution across the two sets does not lead to a performance degradataion
> 3. While PLM and cross-annotation perform similarly in data abundant scenarios, PLM is more accurate in data scarce scenarios. Furthermore, we expect that PLM could lead to a potential solution to minimize error propagation or catastrophic forgetting issues of continual learning (basic idea of the solution is discussed in 1), we left it for the future work.

---

### Meta-Review · Area_Chair_AZom · 2023-09-18

**Recommendation:** 4

**Metareview:**

The paper extends the task of NER with newer categories, named taxonomy expansion. The difficulty is how to train the NER model with two disjoint, partially labeled training sets. To this end, the paper proposed a new approach that employs a modified loss function to capture the relatedness between the original labels with the new ones. The experimental results on six datasets show the proposed methods could cover more domains and types. The paper is clear and easy to follow. The cross-annotation idea is interesting.

---

### Decision · Program_Chairs · 2023-10-07

**Decision:**

Accept-Main

**Comment:**

The paper extends the task of NER with newer categories, named taxonomy expansion. The difficulty is how to train the NER model with two disjoint, partially labeled training sets. To this end, the paper proposed a new approach that employs a modified loss function to capture the relatedness between the original labels with the new ones. The experimental results on six datasets show the proposed methods could cover more domains and types. The paper is clear and easy to follow. The cross-annotation idea is interesting.